# Three-Dimensional Printer-Assisted Electrospinning for Fabricating Intricate Biological Tissue Mimics

**DOI:** 10.3390/nano13222913

**Published:** 2023-11-08

**Authors:** Komal Raje, Keisuke Ohashi, Satoshi Fujita

**Affiliations:** 1Department of Advanced Interdisciplinary Science and Technology, University of Fukui, Fukui 910-8507, Japan; komal@biofiber-fukui.com; 2Department of Frontier Fiber Technology and Sciences, University of Fukui, Fukui 910-8507, Japan; 3Life Science Innovation Center, University of Fukui, Fukui 910-8507, Japan

**Keywords:** three-dimensional printing, electrospinning, three-dimensional nanofiber scaffolds, sacrificial mold, inverse three-dimensional printing

## Abstract

Although regenerative medicine necessitates advanced three-dimensional (3D) scaffolds for organ and tissue applications, creating intricate structures across scales, from nano- to meso-like biological tissues, remains a challenge. Electrospinning of nanofibers offers promise due to its capacity to craft not only the dimensions and surfaces of individual fibers but also intricate attributes, such as anisotropy and porosity, across various materials. In this study, we used a 3D printer to design a mold with polylactic acid for gel modeling. This gel template, which was mounted on a metal wire, facilitated microfiber electrospinning. After spinning, these structures were treated with EDTA to remove the template and were then cleansed and dried, resulting in 3D microfibrous (3DMF) structures, with average fiber diameters of approximately 1 µm on the outer and inner surfaces. Notably, these structures matched their intended design dimensions without distortion or shrinkage, demonstrating the adaptability of this method for various template sizes. The cylindrical structures showed high elasticity and stretchability with an elastic modulus of 6.23 MPa. Furthermore, our method successfully mimicked complex biological tissue structures, such as the inner architecture of the voice box and the hollow partitioned structure of the heart’s tricuspid valve. Achieving specific intricate shapes required multiple spinning sessions and subsequent assemblies. In essence, our approach holds potential for crafting artificial organs and forming the foundational materials for cell culture scaffolds, addressing the challenges of crafting intricate multiscale structures.

## 1. Introduction

Biomimetic structure fabrication plays a pivotal role in cellular biology applications. Cells interact directly with their environment and adapt their behavior accordingly. These dynamic interpretations primarily occur within the extracellular matrix (ECM), a complex network comprising fibrous proteins embedded within a compressible proteoglycan network. The ECM considerably influences cellular behavior and morphology by either inducing or inhibiting specific processes via cell–matrix attachments. Notably, because cells exhibit contrasting growth patterns on planar and fibrous substrates [1], three-dimensional (3D) cultures on biomimetic surfaces offer a closer representation of in vivo conditions.

The ECM of connective tissue is predominantly composed of bundled collagen. Type I collagen, often found in conjunction with type III collagen, constitutes about 90% of the collagen in the mammalian body. It is arranged in fibril bundles of 10–300 nm in diameter and up to a several micrometers in length [2,3]. Type IV collagen, on the other hand, forms the meshwork of the more flexible basement membrane that covers the endothelial and epithelial tissue layers [2,4]. The elasticity of many tissues, such as the arteries and vocal cords, is due to elastic fibers consisting of the ECM protein elastin interspersed with microfibrils [5]. Electrospinning has the capability to produce fibrous interfaces that are beneficial for both endothelial and epithelial cell cultures [6,7]. Although this technique provides essential topographical guidance for the cells, it often results in the formation of two-dimensional (2D) structures or slender meshes. In contrast, 3D printing enables the fabrication of intricate scaffolds with millimeter-scale attributes. Merging these techniques allows the creation of structures with well-defined macroscale and nanoscale features. Given the complex nature of mammalian anatomy, we explored the potential of this combined methodology to manifest intricate geometries from the design phase to realization.

Three-dimensional printing and electrospinning have been extensively integrated into various applications [8]. One prominent method involves electrospraying a polymer-based thin fibrous membrane onto a 3D-printed pattern to produce unique topographical designs [9,10]. These 3D-printed structures not only offer topographical advantages, but also impart notable mechanical strength. They can function as an outer shield for the delicate electrospun layer, forming a vascular tubular graft that drives human mesenchymal stem cell differentiation towards the vascular endothelium [11]. Alternatively, the 3D-printed pattern can function as a sturdy internal frame for an external mesh of electrospun fibers, with applications such as the enhancement of proliferation and modulation of the osteoblast response in bone tissue engineering [12]. Furthermore, bioprinting, a specialized branch of 3D printing, can emulate complex anatomical structures, such as branched tubules. Such structures hold promise for various applications, such as patient-specific grafts tailored to specific surgical procedures, including pediatric congenital heart repairs that often involve constraints related to size and geometry [13].

Manipulating fiber fabrication at a microlevel offers various possibilities in electrospinning. Techniques for fine-tuning electrospinning include altering the electric field using movable pins [14] or modifying the collector surface to be differentially conducting [15]. For example, when designing a hollow 3D electrospun scaffold, it is common to use a rotatable mandrel with a specific diameter that can be easily removed after spinning [11]. Another technique involves the use of a dissolvable core to create voids, either by removing salt crystals [16] or by intercalating 3D-printed structures [17]. Additionally, in this sacrificial core method, it is possible to create a defined surface microtopography [18]. The nature of the electrospun mesh allows for various adaptations, such as converting it into a tube using Teflon sticks of a prescribed diameter [19] or using an ink suitable for 3D printing [20]. These adaptations enable the creation of tailored surface microtopographies. Despite this, all methods thus far have a limited control of 3D features, restricted to either the overall shape or simply one 2D surface of a 3D scaffold. This limitation falls short of replicating an intricate natural 3D microenvironment.

Our proposed methodology aims to address these shortcomings by combining 3D printing and electrospinning techniques, thereby providing a broader range of scaffold geometries and facilitating more physiologically relevant cell–scaffold interactions. Drawing inspiration from prior endeavors, where we engineered a nanofiber-mâché ball using an alginate bead [21], we have expanded upon this methodology. We explored the potential of crafting a diverse array of shapes that can elevate the level of macroscale biomimicry. By incorporating an alginate sacrificial core, we have successfully produced hollow electrospun nanofiber scaffolds. These 3D microfibrous (3DMF) scaffold structures merge the macroscale attributes, obtained by crosslinking alginate within a 3D-printed mold, with a distinctive nanoscale architecture inherent to electrospinning.

## 2. Materials and Methods

### 2.1. Materials

Elastollan^®^ thermoplastic polyurethane 1180 A10 was procured from BASF (Ludwigshafen, Germany). Tetrahydrofuran (THF), ethanol (EtOH), paraformaldehyde (PFA), t-butyl alcohol, Dulbecco’s Modified Eagle Medium (DMEM), polyoxyethylene (20) sorbitan monolaurate (Tween 20) and Hoechst 33342 were obtained from FUJIFILM Wako Pure Chemical Corporation (Tokyo, Japan). Sodium alginate (300 cps), calcium chloride dihydrate and Blocking One were procured from Nacalai Tesque Inc. (Kyoto, Japan). White polylactic acid (PLA) filaments were purchased from Creality Ender (Shenzhen, China). Stainless steel wires were procured from Hikari Co., Ltd. (Osaka, Japan). UltraPure™ 0.5M EDTA (pH 8.0) was obtained from Thermo Fisher Scientific (Waltham, MA, USA). Phosphate buffered saline (PBS) was procured from Nissui Pharmaceutical Co., Ltd. (Tokyo, Japan). Alexafluor 488 was obtained from Abcam Inc. (Tokyo, Japan). Fibronectin from human plasma was procured from Sigma-Aldrich (Tokyo, Japan).

### 2.2. Fabrication Process

The CADs for the inverse mold were designed using DesignSpark Mechanical 5.0 (RS DesignSpark, London, UK), sliced with Cura 5.2.1 (Ultimaker B.V., Geldermalsen, Netherlands) and 3D printed from PLA with a grid infill pattern at 25% fill density Anycubic Mega (ANYCUBIC, Shenzhen, China). The PLA molds were submerged in a 10% CaCl_2_ aqueous solution for 2 min. The molds were then removed from the bath, and a 5% alginate solution in distilled water was introduced into the mold cavity, along with a stainless steel wire running through the center of the mold. This setup was then reintroduced into the 10% CaCl_2_ bath. The alginate templates crosslinked around the wires at room temperature (25 °C); the smaller and larger templates were mounted on 0.5 and 1 mm diameter wires, respectively. The crosslinking time depends on the volume of the structure. For example, the small cone crosslinked within 15 min, while the large cone was left overnight. If the mold is opened before the crosslinking process is complete, the alginate template can fall apart. Thus, the exact crosslinking times are not known; however, all structures were well crosslinked within 24 h. The wire, when connected to the ground, can be rotated at a defined speed within the electrospinning apparatus, serving as a grounded collector.

To prepare the spinning solution, PU was dissolved in THF at 12.5% (*w*/*v*). Electrospinning was performed at 29 kV with a solution flow rate of 0.8 mL/h and a needle-to-wire distance of 12.5 cm on a NANON-01B (MECC Co., Ltd., Fukuoka, Japan). During the spinning, the spinneret moved horizontally at a velocity of 10 mm/s (Appendix A), covering a distance slightly greater than the length of the respective scaffold. The rotational speed of the collector was adjusted according to the design. The electrospinning conditions are summarized in Table 1.

The electrospun scaffolds with alginate were submerged in a jar of 0.05 M EDTA aqueous solution and left on a mixing rotor at 21 rpm to chelate calcium ions and remove alginate. While most scaffolds can dissolve alginate overnight, larger scaffolds (volume greater than 12 cm^3^) with smaller holes from the electrospinning wire (diameter 0.5 mm) required up to two days. The hollow scaffolds were dried at room temperature overnight, at 60 °C for 2 h, or at 120 °C for 15 min.

### 2.3. Mechanical Analysis

Tensile testing was performed on a hollow 3DMF cylinder using Force Tester MCT (A&D Company, Limited, Tokyo, Japan) with a load cell of 500 N. The cylinder was 25.08 mm long on loading and its inner and outer diameters were 5.55 mm and 4.15 mm, respectively. Stress–strain data were normalized to the cross-sectional area of the actual material, excluding the inner hollow region. This was calculated as the difference between the cross-sectional areas obtained from the inner and outer diameters.

### 2.4. FTIR Analysis

Fourier transform infrared spectroscopy was performed using Nicolet 6700 (Thermo Scientific). This was completed to confirm the removal of alginate and EDTA from the PU final scaffold.

### 2.5. Cell Culture Study

Prior to use in cell culture, a hollow 3DMF cylinder (length: 5 mm, inner diameter: 1.6 mm) underwent EDTA treatment to remove alginate. It was then washed with 70% ethanol (EtOH) for sterilization. The process involved submersion in an EtOH bath for 10 min, interspersed with an EtOH rinse of the inner layer of the cylinder via a syringe. Following this, the EtOH-treated cylinder was rinsed with sterilized PBS in a similar manner. The inner surface of the cylinder was then exposed to 1 mL of fibronectin solution (10 µg/mL) and incubated at 37 °C for 30 min to allow physical adsorption. Afterward, the scaffold was rinsed with PBS followed by DMEM/10% FBS/1%PS. A HEK293 cell suspension (650 µL), containing 520 × 10^4^ cells, was gently inoculated into the interior of the scaffold using a syringe. The scaffold was then incubated in a tissue culture plate (TCP), submerged in media, and maintained in static culture for 3 days.

### 2.6. Fluorescence Microscopy

The scaffold was longitudinally sectioned under liquid nitrogen. Subsamples from both inner and outer layers were fixed using 4% PFA. The specimens were then treated with Tween 20 and Blocking One. Subsequently, they were stained with Alexafluor 488 (green) and Hoechst 33342 (blue) and kept in the dark for 2 h. Microscopic observations were carried out using the Olympus IX-81 microscope (Olympus Optical Co., Ltd. Tokyo, Japan).

### 2.7. Scanning Electron Microscopy

The morphologies of the inner and outer layers of the scaffolds were observed using a desktop scanning electron microscope (SEM), JCM-6000Plus (JEOL, Tokyo, Japan) after sputter coating with osmium (MSP-1S, Vacuum Device, Ibaraki, Japan). Initially, square scaffold pieces (5 mm) were sectioned under liquid nitrogen. This step was essential to mitigate the challenge of soft PU fiber distortion during cutting at room temperature. The sectioned samples were mounted on SEM sample holders without any further treatment. The fiber diameters were analyzed using the image processing software Fiji. Measurements were recorded for approximately 100 fibers per image, with the resulting data presented as a frequency histogram.

For the cell culture scaffold, after PFA fixation on day 3, a dehydration protocol was implemented. It involved a gradual substitution with EtOH and an incubation period of 10 min. Following this, EtOH was replaced with t-butyl alcohol in six steps until approximately 100% t-butyl alcohol concentration was achieved. The samples were then refrigerated, followed by freezing. Subsequently, freeze drying was conducted using ES-2030 (Hitachi, Tokyo, Japan).

## 3. Results

In this study, an alginate hydrogel template was constructed by infusing an alginate solution into a 3D-printed mold. Gelation was facilitated by the presence of calcium ions. The PU microfibers were deposited onto this template with electrospinning, resulting in a 3D construct. Subsequently, the alginate template was dissolved in an EDTA solution, yielding a unique 3DMF structure composed exclusively of PU microfibers (Figure 1).

Previously, our colleagues had created a nanofiber-mâché ball from electrospinning on an alginate bead [21]. In this study, we have taken this research further to explore the suitability of this method for different shapes and complex geometries to allow for improved macroscale biomimicry. An alginate sacrificial core was used to produce hollow electrospun nanofiber scaffolds. Furthermore, 3D macroscale features were imparted to alginate by crosslinking it inside a 3D-printed inverse design mold. The hollow 3DMF scaffold preserved both the macroscale features defined with 3D printing and the microscale architectural characteristics of electrospinning.

Cone molds with three distinct dimensions were fabricated using 3D printing. These molds were subsequently used to fabricate 3DMF scaffolds (Figure 2). These rudimentary structures consistently manifested a uniform shape, highlighted by a small aperture at the juncture where the alginate was perforated to accommodate the wire of the electrospinning collector. See the videos for a demonstration of electrospinning in action (Appendix A).

Similarly, cylindrical 3DMF scaffolds with different lengths and diameters were fabricated. Simple structures were easily produced and had a consistent shape, with a small opening at the location where the alginate was pierced with the wire, which was used as the electrospinning collector. Modification of the cylinder resulted in a bifurcating tubular structure (Figure 3c, Appendix A). The Y-tube scaffold can be spun in two ways, by rotating the template along the shorter or longer axis. The orthogonally intersecting 3DMF tube structure (Figure 3b, Appendix A) was designed with consideration that applications involving fluid flow, such as cell culture under flow, can have perpendicular channels that differ from a forked tube with unidirectional flow.

Scaffold designs that were initially found suitable for curved geometries prompted the assessment of alternative structures for linear and planar configurations. A multifaceted scaffold was designed to ascertain its compatibility with the sharp-edged geometries (Figure 3a, Appendix A). This template is characterized by multiple corners, a jagged external surface, and a narrow central segment that anchors its two more-substantial ends. An irregular deposition pattern of microfibrous membranes was observed. There was a pronounced susceptibility to tearing, especially at the sharp corners.

SEM investigations confirmed that both the internal and external surfaces were uniformly covered. The microfibers exhibited a random orientation and significant diameter variations (Figure 4). The thickness of the scaffold was 0.23 mm. Both the outer (0.97 ± 0.63 µm) and inner (0.96 ± 0.61 µm) layers of the scaffold showed fibers with comparable average diameters and standard deviations, indicating no obvious architectural differences between the two sides (Figure 4b). The high standard deviation with respect to the mean indicates a wide range in fiber diameters, with most lying between 350 nm and 1.5 µm. Such non-uniformity among fibers within the same electrospun scaffold is characteristic of electrospinning. This can be ameliorated by enhancing the stability of the Taylor cone, which is influenced by the applied voltage or by adjusting the needle-to-collector distance.

Next, a hollow 3DMF cylinder scaffold, designed to mimic a blood vessel, was fabricated. The cylinder was 25.08 mm long on loading and its inner and outer diameters were 5.55 mm and 4.15 mm, respectively. Its tensile strength was based on the breaking point. The elastic modulus was determined from the initial linear region of the stress–strain curve, up to a 50% strain. During the tensile test, the cylinder exhibited an elastic modulus of 6.23 MPa, and achieved an elongation of 206% at a stress of 0.79 MPa. It then detached from the testing grip holder after undergoing an irreversible length deformation, changing from 25 mm to 29 mm, before any further stretching could take place (Figure 5). Although it did not tear, it failed to revert to its original length even after several days, indicating plastic deformation.

The FTIR analysis compared alginate, PU and the washed PU scaffold. The plot for the final PU scaffold closely aligned with the PU plot, distinguishing it from the alginate plot (Figure 6). This comparison indicated that the alginate was effectively removed post-EDTA processing, and the subsequent EtOH wash ensured the removal of any residual EDTA (Figure 6). The absence of alginate is crucial for cell culture studies, as its presence could alter cell adhesion. Similarly, any remaining EDTA might be cytotoxic. While THF possesses significant toxicity, its high volatility allows for rapid evaporation during electrospinning. The EDTA and EtOH washes further ensure the complete removal of any residues, making the scaffold suitable for biological applications.

Next, a cell culture study was conducted to assess cell adhesion and the non-toxicity of the scaffold to cells. Fluorescence microscopy images from the cell culture study showed that cells were able to attach to the inner layer of the 3DMF cylinder (Figure 7a–f). These images were notably bright due to the autofluorescence from the PU material. SEM observations also revealed the debris of cells on the inner layer of the cylinder, which was absent from the outer layer (Figure 7g–l). This confirmed that cells seeded on the inner layer did not penetrate to the outside of the scaffold.

Finally, scaffolds with complex geometries and multiple shapes were produced. The voice box model has a core feature that distinguishes it from other models: two consecutive conical segments leading to a unidirectional streamlined taper (Figure 8a, Appendix A). This configuration shows a gradual transition along the cone’s gradient while maintaining distinct edges at the cone’s bases. The film had an average thickness of 3.46 mm with uniformly spun fibers.

The scaffold inspired by the tricuspid valve adopted a trifurcated cylindrical design augmented with three equidistant internal supports reinforcing the primary cylinder (Figure 8b, Appendix A). The fabrication methodology involved the creation of three circular sector prisms, each encompassing 120°, which were subsequently amalgamated into a single cylinder. Instead of bonding the components, they were assembled in a rotating mandrel collector. The exterior was solidified via electrospinning. By leveraging this dual-stage electrospinning method, intricate geometries with detailed internal arrangements can be realized.

## 4. Discussion

Conventional electrospinning is restricted by the size, shape and positioning of the collector and often produces 2D sheets or thin meshes that are not representative of the in vivo 3D microenvironment. While electrospinning is somewhat restricted in the macroscale morphologies that it can produce, 3D printing generally does not produce ECM mimicry at a microscale. An exception is bioink printing or bioprinting, which allows the printing of biocompatible materials at higher resolutions [22]. However, this method requires advanced printing equipment, which is expensive. Additionally, printed scaffolds are restricted to soft biomaterials, such as alginate, agarose, cellulose and ECM. Thus, they are confined to a narrow range of mechanical properties and can be used to support various tissue types.

Conversely, the 3D printing of polymers opens the door to fine-tuned mechanical properties. Fused deposition modeling (FDM) is performed at high temperatures to melt polymers so that they can be properly extruded in a layer-by-layer manner [23]. This makes the method unsuitable for the immediate incorporation of cells. Large nozzles (a few millimeters wide) also prevent the printing of high-resolution filaments or fibers. In general, FDM-printed polymer structures are used as support structures.

PU, a flexible bioinert polymer, is compatible with both electrospinning and 3D printing methodologies. In particular, elastic PU has been employed to craft tissue engineering scaffolds, such as 3D printing a water-based ink of PU and other bioactive ingredients suspended in hyaluronan [24]. In this study, PU was used to synthesize 3D structures with microfibrous layouts (Table 1). The elasticity of PU is highly beneficial in forming stretchable membranes. This feature was utilized in our study to form soft elastic PU sheets that wrapped around a 3D structure, with the elastic sheets maintaining this form even after the removal of the sacrificial core. Consequently, electrospinning ensures fibrous architecture at a microscale level, with a potential to mimic ECM [25].

Most approaches for combining 3D printing and electrospinning utilize 3D printing to form networks or meshes. Some approaches use a 3D-printed pattern to transfer to an electrospun membrane, such as embossing, to increase the surface area and improve the efficiency of dialysis [26]. The underlying pattern can also be produced with other methods such as lithography, which has a higher resolution (a few micrometers) but requires specific equipment and expertise for fabrication [27]. Three-dimensional printing can also be used to directly fabricate a rotating mandrel with defined structures (such as branches) [28]. Although this may be useful when the 3D-printed material is sufficiently biocompatible and provides mechanical support, it may not be suitable for all applications. Although multiple approaches have been attempted regarding the combination of electrospinning and 3D printing, this research focuses on biomimicry on the inner hollow side of the structure, as opposed to the outer surface biomimicry, or on using electrospinning to support an already prepared 3D structure. The actual 3D-printed mold is not incorporated for biological use, removing the requirements of biocompatibility and non-cytotoxicity, and expanding the options available to define the morphology of materials.

Remarkably, the geometry of conical 3DMFs was preserved across all dimensions, even with the smallest size (Φ 8 mm, h 10 mm) maintaining its exact form (Figure 2). The surface morphology was found to be dependent on the template size. Nuanced protrusions and indentations were evident in the structures derived from the most substantial templates. This was hypothesized to result from the intricate design of the mold mirrored during the layered 3D printing process. When the spinning time remained constant, there was an observable reduction in the thickness of the structure, corresponding to an increase in the template size. In structures fabricated from the largest template, the aforementioned nuances were transferred to a spun microfiber assembly. For templates of varying sizes, although subtle protrusions and indentations of the mold were replicated on the gel template, detailing was not transposed onto the resulting 3D nanofiber matrix. Structures with diverse surface morphologies can be effectively produced by adjusting both the spinning time and template dimensions.

With electrospinning, the lack of control over the exact location of the fiber deposition is unavoidable. In the branched structures, because the fork was on the inner side of the wire and away from the spinneret, some webbing occurred across both arms (Figure 3c). This also occurs when the Y-tube is electrospun asymmetrically. Similarly, a structural anomaly at the neck of the voice box 3DMF scaffold led to fiber bridging during electrospinning, which deviated from the intended design (Figure 8a). Mitigating this issue would necessitate modifications, such as reducing the electrospinning speed (specifically, the injection volume) or decelerating the collector rotation speed. The issue of webbing can be avoided by separately electrospinning forks of the Y-tube and by combining them using a method similar to that used to produce a tricuspid valve-inspired scaffold. While this method requires an additional electrospinning step, it can produce hollow structures with internal branching and no webbing.

While the alginate templates were identical to the mold, the template may easily rupture when pulled out with force, thus affecting the final 3DMF. This can be prevented by using a suitable concentration of alginate (between 5–8% aq.) to ensure some stiffness and by carefully removing it from the mold using minimum force, without pulling on weak tapered edges.

Notably, when fabricating orthogonally intersecting pipes, the two pieces of the mold do not necessarily have the same design; instead, they mirror each other. Being an asymmetrical design, the intersecting pipe required two molds that were mirror images of each other (Figure 3b). Thus, increasing design complexity adds complexity to the two main steps of the fabrication process: CAD and alginate stability.

There are two scenarios in which a microfibrous membrane is deposited unevenly: when structures with pointed edges and sharp corners undergo extreme stretching and pulling. Thus, this makes them highly prone to tearing. Additionally, when electrospinning is performed without horizontal spinneret movement, the fibers are deposited directly under the needle tip with a small circular area around them. If the alginate cast is large (length > 3 cm), the edges will not accumulate any fibers.

To enhance shape precision, it is imperative to address the discrepancies in the blueprints of a 3D printer during mold creation. Despite the absence of such adjustments, the fabricated structures closely resembled the intended blueprints. This suggests that accommodating the inaccuracies of a 3D printer during mold fabrication can potentially yield even greater structural fidelity.

The cylindrical scaffolds fabricated with PU exhibited high stretchability, aligning more with the mechanical performance of elastin (up to 300%), rather than blood vessels, which are the most common cylindrical anatomical features in the human body [29,30]. This suggests that a hollow 3DMF cylinder from PU may be recreated and adjusted to match a specific size of a selected blood vessel for a better comparison of mechanical properties.

Although other polymers might better fit the requirements of a suitable cell culture scaffold, PU is an affordable and easily spinnable material. It can be used repeatedly to explore the method of using sacrificial templates. As the method is further refined and exact 3D-printed designs are determined, scaffolds could be produced from more biologically and mechanically suitable polymers that dissolve in safer electrospinning solvents. For example, a scaffold might be fabricated from mechanically stiffer and biodegradable material, such as PHBH, to compensate for the strength that the collagen component usually contributes. Additionally, attempting to manufacture a composite hollow 3DMF with multiple materials is feasible, provided they match well in terms of electrospinnability and cytocompatibility and do not adversely interact with each other.

Based on the preliminary cell culture study, cells were observed to successfully attach to the inner layer of the 3DMF cylinder (Figure 7a–i). This is likely due to the treatment of fibronectin on the inner layer, as cells did not attach to the outer layer (Figure 7g–l). Given that electrospinning typically produces scaffolds with small impenetrable pores, it is suitable for applications where a boundary is needed, and where cell migration has to be inhibited or restricted to a specific area. The cell sheet may have been pulled apart and torn when the cylinder was unfurled and flattened for SEM imaging, indicated by the fractured islands. This could also result from the freeze drying process. However, given the successful attachment of cells, further exploration with different shapes and larger diameter cylinders should be tested to better mimic anatomy and facilitate improved imaging. The high autofluorescence from thermoplastic PU posed challenges for fluorescence imaging. This can be considered as a limitation of the specific PU used and suggests the need for a more suitable polymer [31]. Beyond changing the polymer, alternative imaging techniques might be used, such as using molecular probes or reporter-labeled fluorescent cell lines [32,33].

While the immediate next step involves establishing a long-term cytocompatibility study, the controllability of the area of cellular growth presents a broader potential in multicellular or layered multiscale tissue-engineered constructs. In this context, restricting cells might be essential for proper tissue development or regeneration. One such possibility with the current scaffold would be the culture of different cell types on the inner and outer layers of the hollow scaffold, while preserving its 3D macrostructure. Introducing multiple electrospinning steps and cell electrospinning might enable the addition of more layers in an additive process. However, this could limit the ability to manipulate an innermost layer once outer layers are deposited around it. Nevertheless, the presented method could be further developed to create macroscale dimension-specific 3D scaffolds for use in 3D cell culture studies, under varying flow profiles and pressures, acting as prototypes for developing implants or conducting drug response studies.

## 5. Conclusions

In the realm of biomaterials, electrospinning has gained prominence for its capacity to fabricate nanofibers by applying high voltages to polymer solutions, subsequently depositing them onto a collector. This method produces anisotropic nanofiber assemblies, paralleling the architecture of the ECM and thus emerging as a promising scaffold for cellular cultures. The versatility of the selected polymer further broadens its applicability in diverse applications. Notwithstanding their advantages, the inherent 2D sheet-like morphology of electrospun nanofibers presents a conundrum in terms of mechanical robustness. These sheets are predisposed to fracturing or seaming when shaped into intricate 3D configurations.

To circumvent this limitation, this study amalgamated 3D printing technology with conventional electrospinning by integrating an alginate gel into the process. This augmented approach has demonstrated proficiency in replicating sophisticated 3D morphologies reminiscent of biological entities, such as internal body structures and the tricuspid valve of the heart, which remain challenging with gel-only techniques. The novel fabrication method based on a sacrificial template produced from a 3D-printed mold was explored to fabricate various hollow structures, maintaining control over macroscale geometry without sacrificing the microscale architecture characteristics of electrospun membranes. As the advantages of both the methods of electrospinning and 3D printing were evident in the final scaffolds, which would not have been possible with either method used standalone, this new fabrication style can be further studied with novel materials for tissue engineering applications requiring specific shape and size constraints.

## Figures and Tables

**Figure 1 nanomaterials-13-02913-f001:**
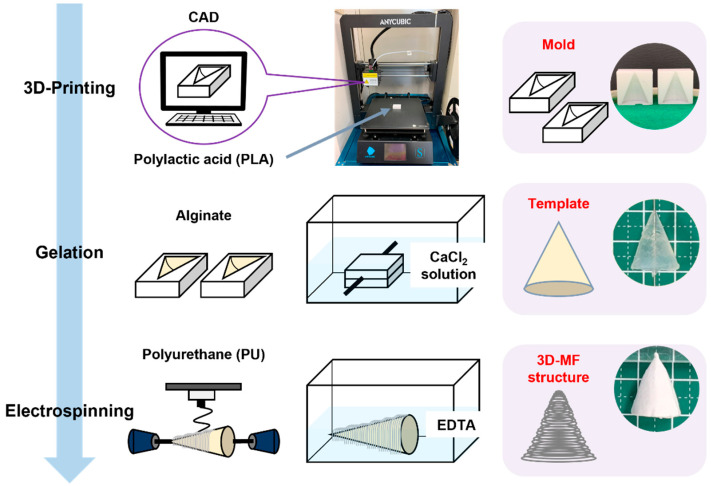
Schematic illustrating the fabrication of an electrospun 3DMF scaffold from top to bottom, with an example of a conical 3DMF scaffold shown on the right. Alginate is crosslinked into the 3D-printed inverse mold made out of PLA to form a template. The template is then used as a rotating collector for electrospinning. Subsequently, it is dissolved in the EDTA solution to obtain a hollow 3DMF PU scaffold.

**Figure 2 nanomaterials-13-02913-f002:**
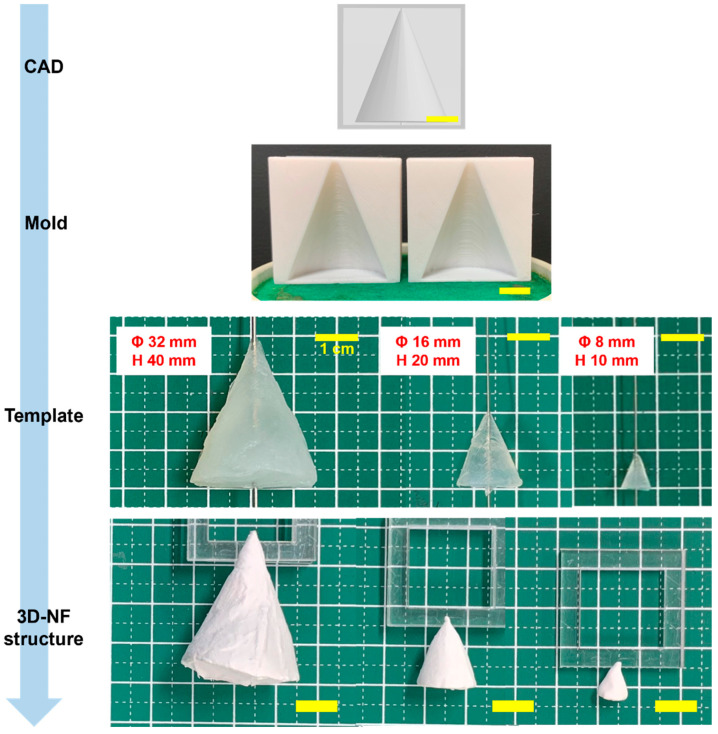
Conical 3DMF scaffolds of various sizes, showing the adaptability of the method for various sizes. The intermediate steps of the fabrication are shown from top to bottom. Scale bar = 1 cm.

**Figure 3 nanomaterials-13-02913-f003:**
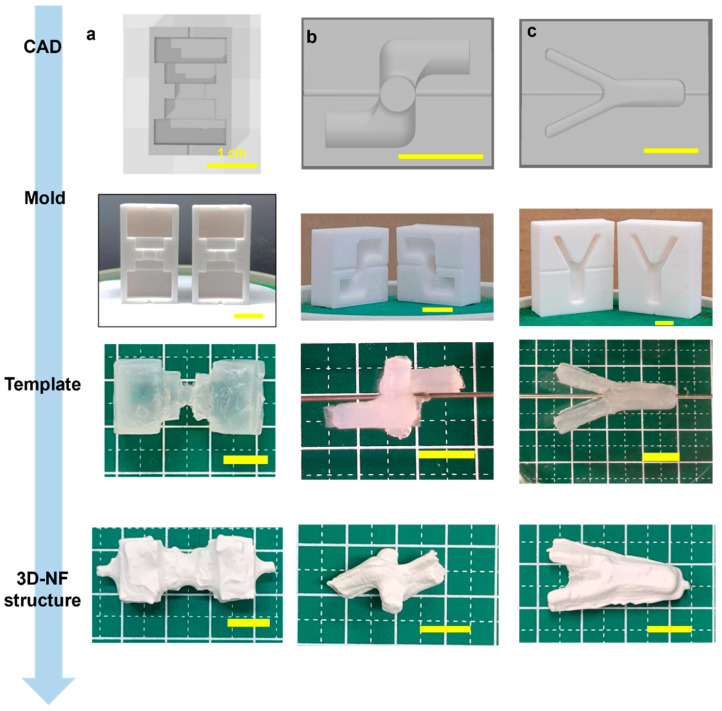
Fabrication of a hollow 3DNF with (**a**) a rugged surface, to test applicability of the method on uneven surfaces, and (**b**) intersecting macroscale features and (**c**) a branched structure, to test the capacity to produce structures with substantial features in axes other than the primary rotation axis.

**Figure 4 nanomaterials-13-02913-f004:**
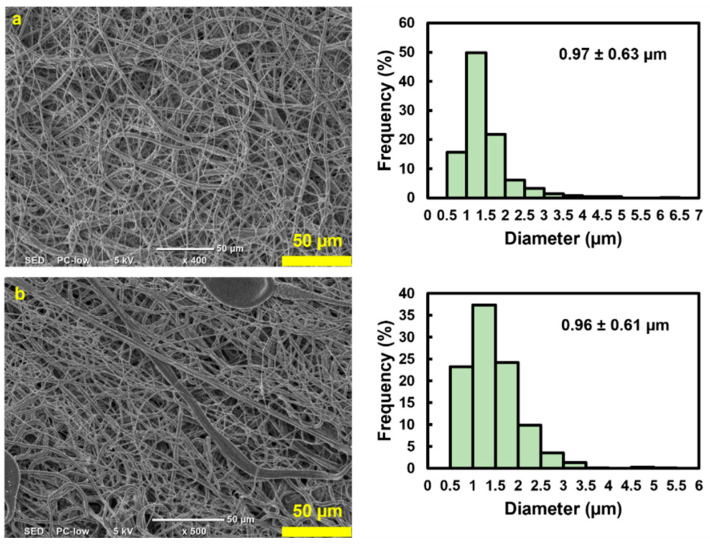
SEM images and the average fiber diameter accompanied with the standard deviation of the (**a**) outer surface and (**b**) inner surface of the rugged structure 3DMF, confirming fibrous architecture on both sides.

**Figure 5 nanomaterials-13-02913-f005:**
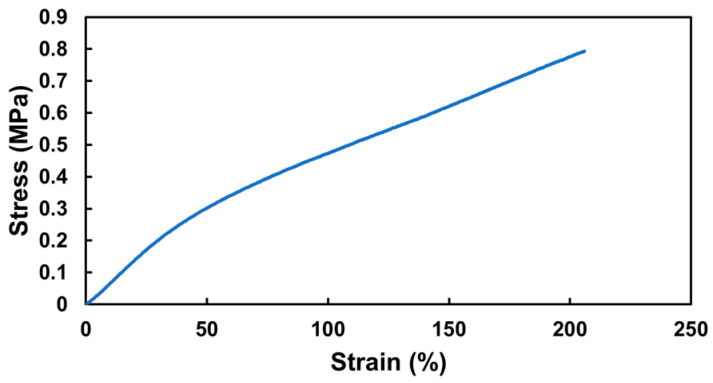
Mechanical performance of hollow 3DMF cylinder scaffold made from PU.

**Figure 6 nanomaterials-13-02913-f006:**
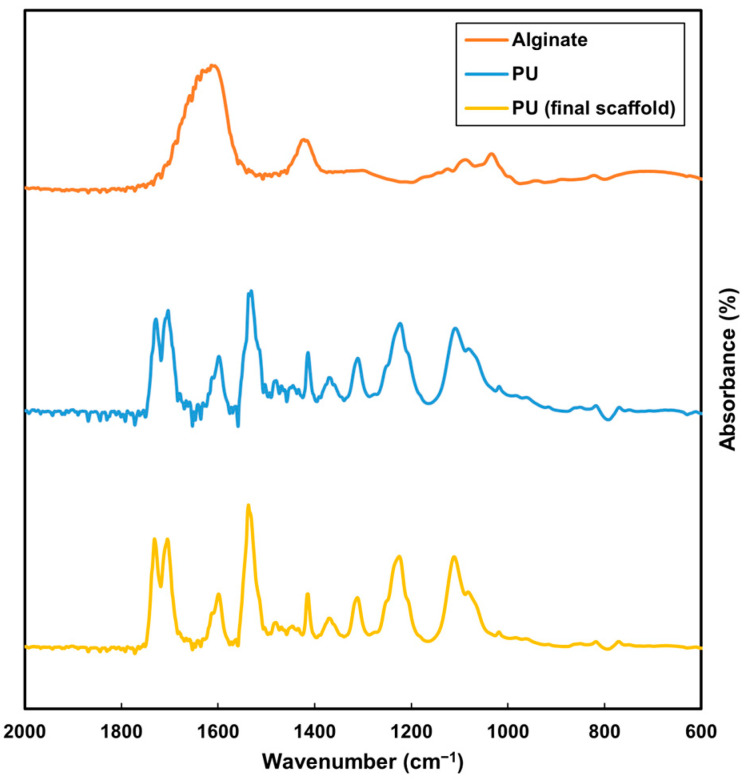
FTIR analysis of alginate (orange), original PU (blue) and washed PU scaffold (yellow).

**Figure 7 nanomaterials-13-02913-f007:**
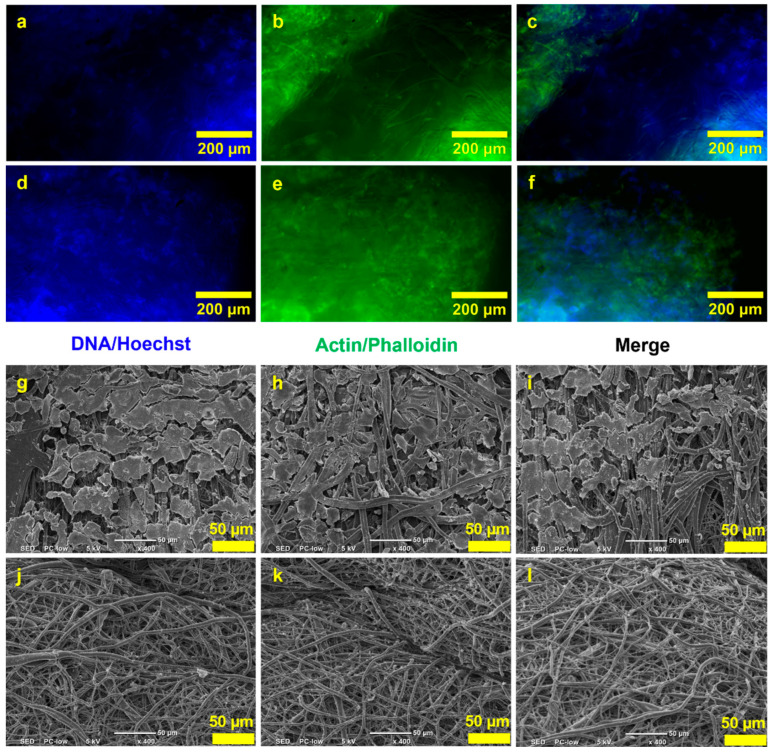
HEK293 culture on the hollow 3DMF cylinder on day 3. Fluorescence microscopy images show the presence of (**a**,**d**) nuclei and (**b**,**e**) actin stained on the inner layers, and (**c**,**f**) composite images. SEM images of (**g**–**i**) the inner layers of the scaffold and (**j**–**l**) the outer layers of the scaffold.

**Figure 8 nanomaterials-13-02913-f008:**
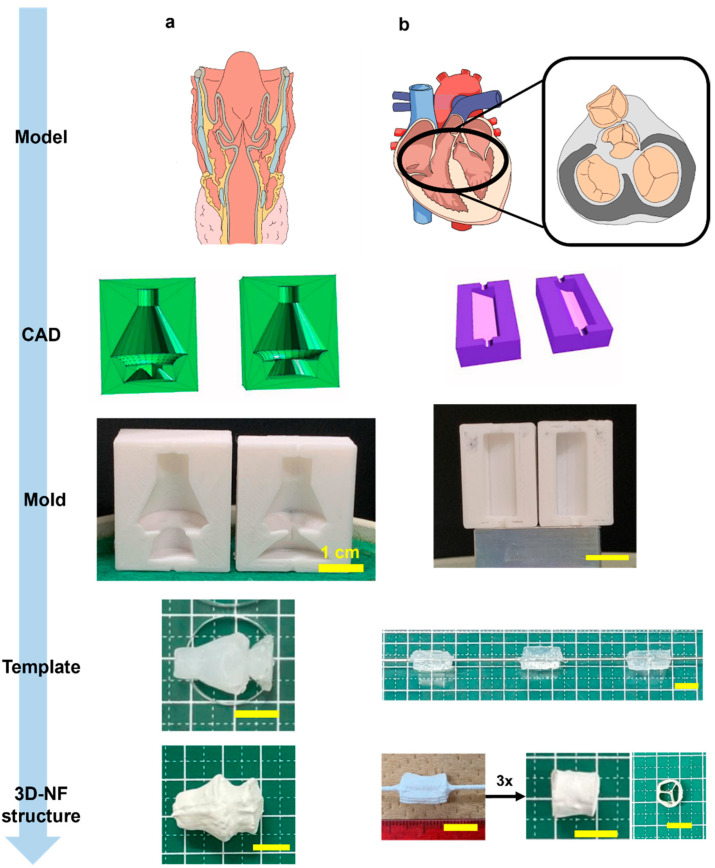
Illustration of 3DMF scaffolds based on anatomical structures. (**a**) Voice box; (**b**) tricuspid valve of heart. The final structures indicate applicability to more complex designs reminiscent of anatomy.

**Table 1 nanomaterials-13-02913-t001:** A summary of electrospinning conditions for various hollow 3DMF scaffolds.

Shape		Spinning Time (min)	Rotation Speed (rpm)	Dimensions (mm)		
**Cones**		90	300	Diameter		Length
	**Small**			32		40
	**Medium**			16		20
	**Large**			8		10
**Rugged structure**		90	300	Total Length	Width	Height
				30	10	16
**Intersecting cylinder**		75	50	Diameter	Length	Width
				5	25	15
**Y-tube** **(both orientations)**		60	50	Diameter		Length
				Stem	Fork	30
				6	3	
**Voice box**		90	300	Diameter		Length
				15.5		24.1
**Tricuspid valve-inspired design (sector)**		90	300	Length	Radius	Angle
				16	5	120°
**Tricuspid valve-inspired design (combination)**		90	700	Length	Diameter	Angle
				16	10	360°

## Data Availability

The datasets generated and/or analyzed in this study are available from the corresponding author upon reasonable request.

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
