# Peer review of "Three-Dimensional Printer-Assisted Electrospinning for Fabricating Intricate Biological Tissue Mimics"

_nanomaterials, 2023, doi:10.3390/nano13222913_

Round 1

Reviewer 1 Report

Comments and Suggestions for Authors

The authors describe a novel methodology that combine 3D printing and electrospinning to manufacture scaffolds. Some concerns should be addressed before publication:

- In Figure 3 and Figure 1 some Japanese characters appear, it must be corrected. Authors should also improve the captions of the figures, as they are poorly described.

- Authors claim potential use of these materials as scaffolds for tissue engineering, however, they haven't shown any experiments that support this hypothesis: are the scaffolds biocompatible for example?

-In line with this question. The scaffolds are made of PU which was dissolved in THF before its electrospinning process. Afterward, the whole structure with the alginate and the electrospun material is immersed in EDTA, which could potentially be cytotoxic if it's free inside the body. Therefore, it would be crucial that the authors assess at least the biocompatibility before concluding the work.

- I am concerned that the methodology here presented is only suitable for manufacturing structures from the edge as it cannot penetrate longer in a structure, limiting importantly the scope of the technique. For example, in Figure 3c, the valvule does not present performance enough to make the "Y-tube" of the artery. Have the authors tried non-symmetric structures? Or, have the authors tried more complex structures?

Author Response

Reviewer 1

The authors describe a novel methodology that combine 3D printing and electrospinning to manufacture scaffolds. Some concerns should be addressed before publication:

- In Figure 3 and Figure 1 some Japanese characters appear, it must be corrected. Authors should also improve the captions of the figures, as they are poorly described.

We apologize for the oversight. We have now made the figure captions more descriptive and translated them into English. Furthermore, we removed the reference fields to prevent auto-correction by software.

- Authors claim potential use of these materials as scaffolds for tissue engineering, however, they haven't shown any experiments that support this hypothesis: are the scaffolds biocompatible for example?

We thank the reviewer for the suggestion. Data from a preliminary cell culture study have been introduced, specifically:

  • 2.1. Materials (L.98-100, 105-108)
  • 2.5. Cell Culture Study (L.149-160)
  • 2.6. Fluorescence Microscopy (L.161-166)
  • 2.7. Scanning Electron Microscopy (L.178-183)
  • 3. Results (L.272-278)
  • Figure 7 (L.279-283)
  • 4. Discussion (L.403-416)

-In line with this question. The scaffolds are made of PU which was dissolved in THF before its electrospinning process. Afterward, the whole structure with the alginate and the electrospun material is immersed in EDTA, which could potentially be cytotoxic if it's free inside the body. Therefore, it would be crucial that the authors assess at least the biocompatibility before concluding the work.

We thank the reviewer for the comment. We performed FTIR analysis to confirm the absence of EDTA, and the following changes have been made:

  • 2.4. FTIR Analysis (L.146-148)
  • 3. Results (L. 261-269)
  • Figure 6 (L.270-271)

- I am concerned that the methodology here presented is only suitable for manufacturing structures from the edge as it cannot penetrate longer in a structure, limiting importantly the scope of the technique. For example, in Figure 3c, the valvule does not present performance enough to make the "Y-tube" of the artery. Have the authors tried non-symmetric structures? Or, have the authors tried more complex structures?

We appreciate the reviewer for pointing out this concern.

We conducted non-symmetrical electrospinning experiments with the Y-tube by positioning it on its shorter axis on the wire, and the result was consistent with the symmetrical mounting shown in Figure 3c. For the orthogonal intersecting pipe structure depicted in Figure 3b, a non-symmetrical CAD approach was employed. The tricuspid valve-inspired design represents the most intricate structure we have fabricated to date, given its capability to support non-webbed internal branching within a hollow structure.

We have incorporated the following modifications in the manuscript to address these points:

  • 4. Discussion (L.358, 362-366)

Reviewer 2 Report

Comments and Suggestions for Authors

Comments on the Quality of English Language

Acceptable.  Some figure references did not use English characters. 

Author Response

Reviewer 2

Review of “3D-Printer Assisted Electrospinning for Fabricating Intricate Tissue Structures”

The authors present electrospinning of 3D structures (cones, Y-tubes, etc.) to mimic the inner architecture of the vocal chords and tricuspid valve. To achieve the 3D shapes, molds were 3D printed and filled with alginate crosslinked with calcium ions. Polyurethane fibers were electrospun onto the mold (rotating the mold continuously during spinning. Following electrospinning, the gel is removed with EDTA leaving the

3D electrospun structure. The paper lacks quantitative results.

Specific Points

The abstract should contain quantitative results.

We thank the reviewer for the suggestion. We have incorporated the quantitative results into the abstract. (L.19, 21-22).

The introduction should better highlight the unique aspects of the process/products presented in the context of existing related work recently reviewed:

Yang, D.L., Faraz, F., Wang, J.X. and Radacsi, N., 2022. Combination of 3D printing and

electrospinning techniques for biofabrication. Advanced Materials Technologies, 7(7), p.2101309.

Thank you for the recommendation. We have added the review paper in the introduction section accordingly. (L.58).

  • References (Ref. 8)

The authors claim that the final 3D structure is “composed exclusively of PU fibers”, but provide no analysis to support their claim.

We appreciate the reviewer's insightful comment. We conducted FTIR analysis to ascertain that the composition of the final PU scaffold is consistent with the PU fibers that were not contaminated with either alginate or EDTA. We have implemented the following revisions to address this:

  • 2.4. FTIR Analysis (L.146-148)
  • 3. Results (L.261-269)
  • Figure 6 (L.271)

Cross sections of the samples made in Figure 2 and Figure 3 would better demonstrate the process and the 3D structures achieved.

We thank the reviewer for the suggestion. We have included supplementary videos to better illustrate the 3D structure. We believe video data adequately conveys the 3D characteristics of the structure. (L.453-455)

  • Video S2-S9

It is unclear what the utility of a nanofiber tricuspid valve would be as the mechanical properties of electrospun materials are relatively weak.

We appreciate the reviewer's comment. The tricuspid valve-inspired design was pursued to assess our method's availability in generating complex macroscale geometries. Exploring experiments with more robust materials remains an avenue for future work.

Figure 1 should better match the process shown in the Video S1 (i.e. the template is spun not the mold)

We thank the reviewer for the suggestion. We have revised Figure 1 to depict the electrospinning of a cone, which more effectively emphasizes the distinction between a template and a mold . Furthermore, we have reviewed the consistent usage of the terms “template” and “mold” throughout the manuscript.

The methods section does not clearly describe the process. There are materials listed, but not how they are used. For example, white polylactic acid (PLA), stainless steel wires.

We sincerely apologize for the oversight. We have made appropriate revisions to the methods section to address this.

  • 2.2. Fabrication Process (L. 111-123)

For the electrospinning setup, the distance that the spinneret moved should be indicated.

The dimensions of the shapes should be included in Table 1

We appreciate the reviewer's thoughtful suggestion. Given that the spinneret movement distance aligns with the shape length, we have updated the methods section (L. 128) and the corresponding table accordingly.

  • Table 1

The SEM sample preparation should be described in more detail. Was the sample imaged whole? If not, how was it sectioned/cut?

We appreciate the reviewer's keen observation. We have updated Methods section.

  • 2.7. Scanning Electron Microscopy (L. 168-183)

Figure references in the text should be in English (e.g lines 139-140)

We apologize for the oversight. We have now made the figure captions more descriptive and translated them into English. Furthermore, we removed the reference fields to prevent auto-correction by software.

Reviewer 3 Report

Comments and Suggestions for Authors

This ms. describes the elaboration of 3D scaffolds in bioinert thermoplastic polyurethan (TPU) mimicking complex biological structures and their characterization at the macroscopic and microscopic level. The elaboration of such materials is allowed by the combination 3D-printing   and electrospinning techniques.

The original and alternative method proposed in this study overcomes limitations of conventional electrospinning or gel-only techniques, allowing to construct sophisticated 3D scaffolds of various geometries. The multi-step elaboration comprising PLA inverse mold printing, gelation of alginate and electrospinning of TPU is well described. It is clearly demonstrated trough the results and discussion parts the capacity of this protocol to produce anisotropic fibers assemblies at the mesoscopic scale.

So this ms. is original, promoting innovative ways for the development of artificial organs and the elaboration of 3D scaffolds for cell culture in tissular engineering. Nevertheless, some points must be revised.

Page 1, line 34: “ECM is predominantly composed of bundled collagen”; it could be appreciate to remind that fibrillar collagen 1 and 3 are the main collagens of ECM, and that some ECM are also constituted of a significant part of elastin, in peculiar arteries and aortic leaflets.

Elaboration process

Page 1, line 96: it should be specified that the mold is made of PLA, as well as the filling pattern and its %.

The temperature and time for the crosslinking of alginate must also be specified.

SEM characterization

It is not clear for me the number of sample used for the SEM characterization. The distribution is clearly indicated but what is the value accompanying the average of the diameter?

Discussion part

Since THF is an highly toxic solvent, is there an alternative? Even if it is not the subject of this study, what type of post treatment should be done to eliminate this component?

Moreover, the used TPU is bioinert but not biodegradable; it is argued that PCL that is a biodegradable polymer is not recommended since it forms brittle structures.

Could other biopolymers be used instead of TPU with this innovative process?

These points deserve to be discussed in the discussion part to open other perspectives.

Page 8, line 211-212 ; references should be added to confirm the sentence.

Moreover, mechanical properties are mentioned in several places of the ms., but some quantifiable values such as Young’s modulus, mechanical strength and deformation at yield could be added to support the assumptions.

Minor comments

Page 6, Figure is not 1 but 4.

Page 4, line 139 ; page 5, line 158 ; figure captions must be translated in English.

Author Response

Reviewer 3

This ms. describes the elaboration of 3D scaffolds in bioinert thermoplastic polyurethan (TPU) mimicking complex biological structures and their characterization at the macroscopic and microscopic level. The elaboration of such materials is allowed by the combination 3D-printing   and electrospinning techniques.

The original and alternative method proposed in this study overcomes limitations of conventional electrospinning or gel-only techniques, allowing to construct sophisticated 3D scaffolds of various geometries. The multi-step elaboration comprising PLA inverse mold printing, gelation of alginate and electrospinning of TPU is well described. It is clearly demonstrated trough the results and discussion parts the capacity of this protocol to produce anisotropic fibers assemblies at the mesoscopic scale.

So this ms. is original, promoting innovative ways for the development of artificial organs and the elaboration of 3D scaffolds for cell culture in tissular engineering. Nevertheless, some points must be revised.

Page 1, line 34: “ECM is predominantly composed of bundled collagen”; it could be appreciate to remind that fibrillar collagen 1 and 3 are the main collagens of ECM, and that some ECM are also constituted of a significant part of elastin, in peculiar arteries and aortic leaflets.

We thank the reviewer for the suggestion. We have rewritten the introduction to include the reviewer's suggestion.

  • 1. Introduction (L. 41-48)

Elaboration process

Page 1, line 96: it should be specified that the mold is made of PLA, as well as the filling pattern and its %.

We thank the reviewer for the comment. The reviewer's suggested changes have been made to the Methods section (111).

  • 2.2. Fabrication Process (L. 111)

The temperature and time for the crosslinking of alginate must also be specified.

We thank the reviewer for the suggestion. The Methods section has been updated.

  • 2.2. Fabrication Process (L. 116-123).

SEM characterization

It is not clear for me the number of sample used for the SEM characterization. The distribution is clearly indicated but what is the value accompanying the average of the diameter?

We thank the reviewer for the comment.

We have modified the SEM preparation description. (L. 176)

We also expanded upon the average and standard deviations in the Results sectionto improve interpretation of the results. (L. 235-244)

Discussion part

Since THF is an highly toxic solvent, is there an alternative? Even if it is not the subject of this study, what type of post treatment should be done to eliminate this component?

We thank the reviewer for raising the concern.

We have explained the treatment for wash before cell culture use in detail (L. 150-160).

We also have introduced a discussion of THF (L. 265-269)

Moreover, the used TPU is bioinert but not biodegradable; it is argued that PCL that is a biodegradable polymer is not recommended since it forms brittle structures.

Could other biopolymers be used instead of TPU with this innovative process?

We apologize for the insufficient explanation.

The relevant PCL description has been removed from the manuscript, and instead, a discussion of PHBH, another biodegradable polymer, has been added. We have expanded upon the discussion (L. 393-428) and conclusion (L. 444-451) to clarify that the purpose of the research is not to advocate for TPU as the only suitable polymer.

These points deserve to be discussed in the discussion part to open other perspectives.

Page 8, line 211-212 ; references should be added to confirm the sentence.

We thank the reviewer for the suggestion. The sentence has been modified based on the results, and a reference has been introduced to continue the idea (L. 326).

  • Ref. 25.

Moreover, mechanical properties are mentioned in several places of the ms., but some quantifiable values such as Young’s modulus, mechanical strength and deformation at yield could be added to support the assumptions.

We thank the reviewer for their suggestion. Mechanical properties were tested and the following modifications have been made to the manuscript:

  • 2.3. Mechanical Analysis (L. 139-145)
  • 3. Results (L. 249-258)
  • Figure 5 (L. 259-260)
  • 4. Discussion (L. 388-392)

Minor comments

Page 6, Figure is not 1 but 4.

Page 4, line 139 ; page 5, line 158 ; figure captions must be translated in English.

We apologize for the errors. The figure numbers and captions have been corrected.

Round 2

Reviewer 1 Report

Comments and Suggestions for Authors

The authors have adressed all the suggestions, therefore at this stage, the paper is ready for publication.

Best